# Lower Energy Intake among Advanced vs. Early Parkinson’s Disease Patients and Healthy Controls in a Clinical Lunch Setting: A Cross-Sectional Study

**DOI:** 10.3390/nu12072109

**Published:** 2020-07-16

**Authors:** Petter Fagerberg, Lisa Klingelhoefer, Matteo Bottai, Billy Langlet, Konstantinos Kyritsis, Eva Rotter, Heinz Reichmann, Björn Falkenburger, Anastasios Delopoulos, Ioannis Ioakimidis

**Affiliations:** 1Department of Biosciences and Nutrition, Karolinska Institutet, 171 77 Stockholm, Sweden; billy.langlet@ki.se (B.L.); Ioannis.Ioakimidis@ki.se (I.I.); 2Department of Neurology, Technical University Dresden, 01099 Dresden, Germany; lisa.klingelhoefer@uniklinikum-dresden.de (L.K.); eva.rotter@uniklinikum-dresden.de (E.R.); Heinz.Reichmann@uniklinikum-dresden.de (H.R.); bjoern.falkenburger@uniklinikum-dresden.de (B.F.); 3Division of Biostatistics, Institute of Environmental Medicine, Karolinska Institutet, 171 77 Stockholm, Sweden; matteo.bottai@ki.se; 4Electrical and Computer Engineering Department, Aristotle University of Thessaloniki, 541 24 Thessaloniki, Greece; kokirits@mug.ee.auth.gr (K.K.); adelo@eng.auth.gr (A.D.)

**Keywords:** Parkinson’s disease, energy intake, food, monitoring, eating behavior, weight loss, neurodegenerative diseases, malnutrition

## Abstract

Unintentional weight loss has been observed among Parkinson’s disease (PD) patients. Changes in energy intake (EI) and eating behavior, potentially caused by fine motor dysfunction and eating-related symptoms, might contribute to this. The primary aim of this study was to investigate differences in objectively measured EI between groups of healthy controls (HC), early (ESPD) and advanced stage PD patients (ASPD) during a standardized lunch in a clinical setting. The secondary aim was to identify clinical features and eating behavior abnormalities that explain EI differences. All participants (*n* = 23 HC, *n* = 20 ESPD, and *n* = 21 ASPD) went through clinical evaluations and were eating a standardized meal (200 g sausages, 400 g potato salad, 200 g apple purée and 500 mL water) in front of two video cameras. Participants ate freely, and the food was weighed pre- and post-meal to calculate EI (kcal). Multiple linear regression was used to explain group differences in EI. ASPD had a significantly lower EI vs. HC (−162 kcal, *p* < 0.05) and vs. ESPD (−203 kcal, *p* < 0.01) when controlling for sex. The number of spoonfuls, eating problems, dysphagia and upper extremity tremor could explain most (86%) of the lower EI vs. HC, while the first three could explain ~50% vs. ESPD. Food component intake analysis revealed significantly lower potato salad and sausage intakes among ASPD vs. both HC and ESPD, while water intake was lower vs. HC. EI is an important clinical target for PD patients with an increased risk of weight loss. Our results suggest that interventions targeting upper extremity tremor, spoonfuls, dysphagia and eating problems might be clinically useful in the prevention of unintentional weight loss in PD. Since EI was lower in ASPD, EI might be a useful marker of disease progression in PD.

## 1. Introduction

Parkinson’s disease (PD) is a progressive neurodegenerative disorder characterized by the cardinal motor symptoms of rest tremor, brady-/hypokinesia and rigidity next to a PD-specific non-motor profile [1,2,3]. PD affects an estimated 1–2 per 1000 persons of the population and the prevalence increases with age, with approximately 1% of the population above age 60 being affected [4].

The core diagnostic criteria for PD have traditionally been the motor abnormalities. Recently, more focus has been given to non-motor symptoms (NMS), which also have been included in the diagnostic criteria for PD [1,2]. Important and common NMS of PD with respect to nutrition are dysphagia and constipation [5], olfactory and taste impairment [6], depression [7], weight loss and malnutrition [8]. These NMS together with the PD-specific motor symptoms potentially have an effect on eating behavior [9,10]. Weight loss and malnutrition might further worsen the progression of the disease and they have been implicated as potential novel targets for PD interventions [11]. Furthermore, malnutrition has been associated with worse health outcomes such as early mortality, impaired immune function [12], slower wound healing, prolonged hospital stays, impaired bone health and increased frailty [13], as well as negatively affecting quality of life [14,15]. While NMS management is slowly being integrated in PD treatments, they are still not the primary focus of patient management [8,11,16]. 

Different treatment strategies in PD have diverse impacts on body weight. For example, deep brain stimulation (DBS), an effective treatment option for motor complications, has been associated with weight gain among treated PD patients [17]. Similar results have also been observed during the initiation of dopamine replacement therapy [18]. Other studies have shown a negative association between daily levodopa dosage per kg of bodyweight and patients’ body mass index (BMI) [15,19], with weight loss in up to 50% of PD patients [10]. Studies also suggest that weight loss is more likely to occur during the later stages of the disease and that it might be related to severe loss of olfaction [15,20]. Altered protein intake is commonly advised to advanced PD patients in order to improve PD medication effectiveness and to avoid fluctuations [21]. However, this practice might increase the risk of long-term nutrient deficiencies and, in the worst-case scenario, malnutrition and weight loss [21,22].

The mechanisms behind the observed weight changes in PD are not fully known, since no objective study quantifying energy intake has previously been conducted in PD patients. Weight loss occurs when energy expenditure is higher than energy intake [23]. In PD, earlier studies showed that resting energy expenditure was higher among PD patients vs. healthy controls [24,25]. However, a later study that measured total energy expenditure (TEE) with doubly labeled water showed that the observed difference did not result in increased TEE. Instead, it was related to a lower total energy expenditure vs. healthy controls due to lower physical activity among the PD patients [26]. Authors therefore argued that the observed weight loss among PD patients must result from a decreased energy intake, potentially due to factors interfering with eating. These might be dysphagia and motor symptoms causing hand-to-mouth coordination problems [26], as well as reduced smell and taste function [11]. This is in contrast to self-reported data that suggest that PD patients have increased energy intake vs. healthy volunteers [22]. However, self-reported data are often biased when it comes to estimations of energy intake and should always be interpreted with caution [27], especially for groups of older patients with potential cognitive impairments or neuropsychiatric conditions such as PD patients [28]. It should be noted that PD is a heterogeneous disease with a range of different motor and non-motor symptoms caused by alpha-synuclein pathology and the involvement of different neurotransmitter systems [29,30]. The inclusion of different PD stages in studies on energy imbalance is therefore important and results should be interpreted based on clinical features that vary between PD patients [11]. 

The primary aim of the current study was to investigate objectively measured energy intake differences among groups of healthy controls, early and advanced stage PD patients during a standardized lunch, in order to better understand the energy balance equation in PD. The study’s secondary outcomes were: (1) to identify clinical features and eating behavior abnormalities that might explain potential energy intake differences between the groups, as well as (2) meal component intake analysis between the included groups. 

## 2. Materials and Methods 

### 2.1. Study Design

A cross-sectional study design was used to explain group-level differences in single lunch energy intake (kcal) among groups of (1) advanced stage and (2) early stage PD patients, as well as (3) healthy controls. 

### 2.2. Setting

Data collection was performed in a clinical lunch setting at the Department of Neurology of the Technical University Dresden (TUD), Germany. The study included: (i) a standardized meal served in front of two video cameras, and (ii) a medical evaluation (Figure 1) with regard to normal bodily effort, which also included examination of medical history, validated questionnaires and scales about motor and non-motor symptoms of PD, as well as a standardized assessment of olfaction, taste and swallowing. The study protocol was identical for PD patients and healthy controls and was completed in all cases in the outpatient facilities of the Neurological Clinic. All PD patients were assessed during “ON” motor state (e.g., all PD patients performed the study assessments at the time point of the best PD symptom control as possible by intake of anti-Parkinson’s medication [31]) in order to better standardize motor symptoms among the included PD patients. The participants were instructed to avoid eating a late breakfast or lunch before coming to the study.

### 2.3. Participants

All study participants were recruited from the in- and outpatient clinics of the Department of Neurology, at the University Hospital of Dresden, Germany. Potential participants were evaluated based on the predefined inclusion and exclusion criteria (see Appendix A for the participant flowchart). Therefore, only patients with idiopathic PD in early and advanced stages of disease [2,32] and healthy controls were included. Any other forms of PD than idiopathic and any other forms of neurodegenerative disorders, such as dementia, were excluded. PD patients who were treated with an advanced therapy (DBS, apomorphine/duodopa pump) were also excluded. Additionally, participants with any of the following conditions were excluded from the study: (i) any contraindication for oral food intake (e.g., known allergy to the standardized meal), (ii) any accompanying disease-causing dysphagia (e.g., throat cancer), (iii) any known active gastrointestinal, endocrine or malignant disease or (iv) any other disease influencing body composition and nutritional status within five years prior to the participation of the study, (v) known smell and/or taste impairment caused by any other reason than idiopathic PD (e.g., by head trauma), as well as (vi) acute major depression. Six of the 23 healthy controls were partners of the included PD patients. The remaining healthy controls were recruited through the promotion and advertisement of the study (e.g., by flyers spread at TUD, newspaper announcements, and study description at the research webpage of TUD). None of the healthy controls were genetically related to a PD patient. Written informed consent was obtained from all participants before any study procedures were performed. The study was approved by the German ethical review board (EK 75022018) and the Swedish ethical review board approved data handling and analysis in Sweden (DNR: 2018/2423-31/2). All the study procedures were in accordance with the Helsinki Declaration [33]. 

### 2.4. The Standardized Meal

Each study participant ate a standardized meal in a quiet room dedicated to the experiment around their usual lunch time (11:00–15:00) during a weekday (Figure 2a,b). The same room was used for all meals. Each participant was seated at a table in front of two video cameras (GoPro HERO 5 [34], recording at 1920 × 1080 resolution with a rate of 30 frames per second). One camera was placed to the left and one in front of each participant. The camera distance was around 1 m and participants ate their meals alone. Additionally, each participant wore two smartwatches (one was attached to the right wrist and one to the left wrist) in order to collect accelerometer and gyroscope data that were used to develop an automatic bite counter algorithm to allow for the analysis of intake microstructure/mechanics [35,36]. These data are not part of the current analysis and the additional equipment did not otherwise affect the study.

The standardized meal consisted of 200 g pre-heated sausages (solid food), 400 g cold potato salad (semi-solid food), and 200 g apple puree (soft food)—a typical German lunch meal (see Table 1 for the macronutrient composition and Figure 2c for a picture of the food that was served). In addition, a standardized bottle of water (500 mL) was available at the lunch table and each participant could drink water before, during and/or after the meal freely. 

All foods were weighed and served on a standardized plate before the meal started, allowing for the standardization of portion sizes across participants. The video recordings were then started by the responsible researcher and the participants ate their meal without any time restrictions. After the meal, each individual food component left on the plate and the leftover water were weighed again. Weighing measurements before and after the meals allowed for the estimation of energy (kcal) eaten from each food component, as well as grams of water consumed. 

### 2.5. Statistical Methods

Multiple linear regression (IBM SPSS version 25) was used to explain variations in the dependent variable single lunch energy intake (kcal) among the three groups: (1) healthy controls, (2) early PD patients, and (3) advanced PD patients (primary outcome). Individual participant status was coded as a binary variable (e.g., “healthy control”, “early PD” and “advanced PD” vs. “not”). Those were tested together with the potential confounding variables of sex, age, height and bodyweight in the primary outcome model. Sex had a significant effect on the primary outcome model and was therefore included in all regression models to control for confounding. Age, bodyweight and height did not affect the primary outcome model and were therefore excluded from all subsequent models. The assumption of normal distribution was checked by visual inspection of a Normal Q-Q plot, linearity and equal variance by visual inspection of a residual plot, collinearity and multicollinearity by variance inflation factor (VIF) and tolerance statistics, and independence by the Durbin–Watson statistic. Cook’s distance >4/n was used as a threshold for the exclusion of high leverage points (outliers) in the primary regression model and the outliers were also excluded in all subsequent regression models. Additional regression models were conducted to investigate variables that could explain group-level differences in energy intake (exploratory analyses). One variable at a time was added to the primary regression models during this process. To test for group differences in participant characteristics, as well as eating behaviors, multiple regression models (both linear and logistic depending on continuous vs. binary outcome data) were conducted, with sex added as a confounding variable in each model. A *p*-value lower than 0.05 was evaluated as statistically significant.

### 2.6. Data Sources/Measurement 

#### 2.6.1. Weight and Height

The weight (by use of a weight scale) and height (by use of a height scale) of each participant were measured by study personnel before each lunch meal. These measurements also allowed for the calculation of participants’ BMIs.

#### 2.6.2. Energy Intake (kcal)

A kitchen scale was used to weigh all individual food components to the exact gram before each meal and leftovers after the meal. From this procedure, the grams eaten from each individual food component could be calculated and, based on the kcal value from the nutritional declaration (Table 1), the total energy intake (kcal) from each individual food component was calculated. The energy intake from each food component was then summed up and the lunch meal energy intake (kcal) was calculated. A similar procedure was used to calculate water intake during each meal. 

#### 2.6.3. Eating Behaviors

Two GoPro video cameras (one in front and one to the left of the lunch table) were used to record videos of each participant while they were eating, as well as during medical examination. The Observer XT (version 12.5) software was later used to annotate each spoonful and chew that the participants took during their respective meal onto the video. A spoonful was defined as the moment when the food was taken off the plate by the participant and was starting the upwards movement up from the plate [37]. A chew was annotated when a chewing motion had been completed (occlusion) [38]. Meal duration was used to enable the calculation of the eating rate and was calculated as the time between the first and last spoonful taken by each participant during their respective meal. Energy eating rate was calculated as energy intake (kcal) divided by total meal duration (min). Eating rate was calculated as food mass intake (g) divided by total meal duration (min). 

#### 2.6.4. Baseline Characteristics and Parkinson’s Medication

Participants’ general medical history, with a special focus on diagnoses of depression, as well as constipation, defined as having bowel movements less than three times a week, was assessed. PD disease duration was calculated as the time period from the date of PD diagnosis to the date of study assessments. To assess the impact of the specific PD medication, the following variables were included: Daily Levodopa Dose (mg/d): total daily dose of immediate and controlled release Levodopa in mg. Levodopa Equivalent Daily Dose (LEDD) (mg/d): total daily dose of all anti-parkinsonian drugs in mg including Levodopa, catechol-O-methyltransferase (COMT)-inhibitors, dopamine agonists, monoamine oxidase (MAO)-B inhibitors, Amantadine, calculated based on the conversion factor [39]. Daily Levodopa Dose (mg/d)/kg bodyweight ((mg/d)/kg): This was calculated based on the body weight of the individual study participant.

#### 2.6.5. Variables to Assess Motor Symptoms

Motor function was assessed by a specialized neurologist using the Unified Parkinson’s Disease Rating Scale (UPDRS) part III [40], the Abnormal Involuntary Movement Scale (AIMS) [41] and the Hoehn and Yahr stage (H&Y stage) [42], just after the meal, in all study participants. A motor “ON” state during the meal and the assessments was ensured by controlling for the last Levodopa medication intake when applicable. Based on the UPDRS single items, the following variables were calculated to investigate the impact of the different PD-specific cardinal motor symptoms, with a focus on the upper extremities (UE): Tremor UE: Sum of the rest and postural tremor of right and left hand (UPDRS III item 20 and 21), range 0–16. Rigidity UE: Sum of rigidity of right and left UE (UPDRS III item 22), range 0–8. Brady/Hypokinesia UE: sum of finger tapping, hand movements and pro-/supination movements of right and left hand (UPDRS III item 23, 24, 25), range 0–24. 

#### 2.6.6. Variables to Assess Non-Motor Symptoms (NMS)

The following non-motor domains were defined: 

Taste problems: The gustatory testing was based on “Taste Sprays” with the four taste qualities: sweet, sour, salty and bitter. The spray was placed on the middle of the tongue as whole-mouth test [43]. The study participants had to identify the taste from a list of four descriptors: sweet, sour, salty, bitter (multiple forced choice). To obtain an impression of overall gustatory function, the number of correctly identified tastes was summed up for the “Taste Spray Value Objective” index with a possible minimum value of 0 (no answer correct) and a maximum value of 4 (all answers correct). This score resulted in the objective evaluation of gustatory function distinguished in normogeusia (value of 4), hypogeusia (value of 2 and 3) and ageusia (value of 0). If participants scored 0–3, a binary variable categorized them as having taste problems. 

Smell problems: Olfactory function was assessed by a standardized psychophysical olfactory test, the Sniffin’ Sticks’ with odorants being presented on pen-like odor dispensing devices [44]. This comprises the tests for odor threshold (n-butanol, testing by means of a single staircase), odor discrimination (16 pairs of odorants, triple forced choice) and odor identification (16 common odorants, multiple forced choice from four verbal items per test odorant). The sum of the value for threshold, discrimination and identification results in the TDI score (range 1–48) and olfactory abilities could be classified in functional anosmia (<16), hyposmia (16–30), and normosmia (>30) [45]. Based on this, study participants were evaluated objectively to suffer from smell problems (anosmia and hyposmia) or not (normosmia). 

Eating problems: Prepharyngeal, pharyngeal and laryngeal functions were evaluated. For subjective evaluation, study participants were asked whether they experience any problems chewing, any problems with oral transport of food within the mouth or any problems swallowing, to be answered with “yes” or “no”, respectively. If any of these three questions was answered with yes by the participant, the participant was evaluated as having subjective eating problems. 

Dysphagia: For objective evaluation a standardized swallowing assessment in the form of the 3-ounce water swallow test [46,47] and the swallowing of pasty and crumbly food were performed [48,49]. Compensation mechanisms such as chin tuck, head turn, effortful swallow, supraglottic and super-supraglottic swallowing and the Mendelsohn Maneuver were observed during the standardized swallowing assessments [50]. Based on the examination, study participants were evaluated to suffer from dysphagia or not. 

### 2.7. Study Size

Before the study, a power calculation (G*Power 3.1) was conducted for group comparisons of differences in energy intake (kcal) among the three groups with balanced sex ratios. A required sample size of *n* = 20 in each group was estimated in order to detect a clinically relevant group-level difference in energy intake of 20% and an expected standard deviation of 40%. However, the recruitment of balanced sex proportion groups was not feasible in practice, since most accessible PD patients were males and most available healthy controls were females. Therefore, multiple regression was used in order to control for the confounding variable, sex, and post-hoc power calculation for the primary outcome analysis revealed an achieved power of 0.85 given an alpha error probability of 0.05, a total sample size of 62 subjects (two outliers excluded in the primary outcome regression model), 3 predictor variables (sex, early PD and advanced PD), and an effect size of 0.15.

## 3. Results

### 3.1. Group Characteristics

Study participants were categorized into three main groups: (1) healthy controls, (2) patients with PD in early stage of disease (Hoehn and Yahr stage (H&Y) ≤ two and disease duration ≤ 5 years), and (3) patients with PD in advanced stage of disease (H&Y > two and disease duration ≥ 7 years). In total, 23 healthy controls (14 females), 20 early PD patients (seven females) and 21 advanced PD patients (eight females) were included. Table 2 displays characteristics for the included groups of participants. 

There were no significant differences in age, bodyweight, height or BMI between the three groups. As expected, due to the selection criteria for the study, advanced PD patients had a significantly longer duration of disease vs. early PD patients as well as a higher H&Y stage. PD medication variables, as well as motor features of PD, were all significantly higher among early and advanced PD patients vs. healthy controls. Advanced PD patients had significantly higher dosages of PD medications vs. early PD patients as well as significantly higher scores for brady-/hypokinesia UE. Furthermore, advanced PD patients had significantly more smell problems, dysphagia and constipation, in comparison to healthy controls. Early PD patients also had more smell problems vs. healthy controls.

### 3.2. Primary Outcome: Group Differences in Energy Intake

Advanced PD patients had a significantly lower energy intake (−162 kcal) compared to healthy controls when controlling for sex (Table 3). Advanced PD patients also had a significantly lower energy intake (−203 kcal) vs. early PD patients. There was no significant difference in energy intake between early PD patients and healthy controls. 

The energy intake (kcal) among the included groups can be seen in Figure 3.

### 3.3. Secondary Outcome 1: Exploratory Analysis Explaining Group-Level Differences in Energy Intake 

Advanced PD patients vs. healthy controls: additional regression models showed that adding the following variables (one at a time) to model 1 led to a more than 10% reduction in the observed lower energy intake among advanced PD patients vs. healthy controls: (1) number of spoonfuls (advanced PD status B value = −99 kcal), (2) tremor UE score (−127 kcal), (3) eating problems (−136 kcal), and (4) dysphagia (−140 kcal, see Appendix A). Adding all the abovementioned variables to primary outcome model 1 modified the observed difference further (advanced PD status B value = −22 kcal).

Advanced vs. early PD patients: In primary outcome model 2, adding number of spoonfuls (advanced PD status B value = −124 kcal), eating problems (−179 kcal), and dysphagia (−181 kcal) reduced the observed lower energy intake among advanced PD patients vs. early PD patients more than 10% (see Appendix A). Adding all the abovementioned variables together into primary outcome model 2 reduced the observed group-level difference further (advanced PD status B value = −100 kcal). 

The following parameters of our study could not explain the reduction in the observed lower energy intake, neither in advanced PD vs. healthy controls nor in advanced PD vs. early PD: BMI, variables of PD medications, the PD motor symptom UE rigidity and non-motor symptoms taste problems, or depression and further variables of eating behavior (e.g., meal duration and eating rate). However, UE brady-/hypokinesia, water intake, constipation and smell problems made the observed difference in energy intake between advanced PD patients vs. healthy controls >10% greater, while constipation did the same when comparing advanced PD patients vs. early PD patients (see Appendix A). 

### 3.4. Secondary Outcome 2: Meal Component Intake Analysis

Meal component intake analysis showed that potato salad, sausage and water intakes were all significantly lower among advanced PD patients vs. healthy controls, while potato salad and sausage intakes were significantly lower among advanced vs. early PD patients (Table 4 and Figure 4). Apple mash intake was not significantly different between the groups. Additionally, advanced PD patients had significantly lower number of spoonfuls vs. early PD patients. There were no other group-level differences in eating behavior between the groups (Table 4). 

## 4. Discussion

This is the first study to objectively quantify energy intake among groups of early and advanced PD patients. Results from our primary outcome analysis showed that advanced but not early PD patients had a significantly lower energy intake vs. healthy controls. The advanced PD patients also had a significantly lower energy intake vs. early PD patients. Our secondary outcome analyses showed that the lower energy intake among advanced PD patients vs. healthy controls could be explained by: (a) higher UE tremor scores (PD motor symptom), (b) increased subjectively reported eating problems, as well as dysphagia (PD NMS), and by (c) taking fewer spoonfuls during the meal (eating behavior variable). In fact, UE tremor, number of spoonfuls, subjective eating problems and dysphagia could explain most (~86%) of the observed difference in energy intake between advanced PD patients and healthy controls. Additionally, eating problems, dysphagia and number of spoonfuls could explain approximately half of the observed lower energy intake among advanced vs. early PD patients. Furthermore, advanced PD patients, vs. both healthy controls and early PD patients, had significantly lower intake of solid food (sausages) and semi-solid food (potato salad), while soft food (apple mash) intake was not significantly different between the groups. Lastly, water intake was significantly lower among advanced PD patients vs. healthy controls.

In the current study, patients diagnosed with PD have been divided into either early or advanced stage of the disease course. This was defined based on disease duration and H&Y stage. H&Y is a commonly used and worldwide accepted scale to describe PD severity and progression, mainly based on motor symptoms and their impact on disability [42]. At H&Y stage 3, an impairment of postural reflexes is requested, which is a sign of an advanced PD stage. Therefore, postural instability is not part of the MDS-PD diagnostic criteria anymore but is labeled as a red flag if recurrent (>1/year) falls appear because of impaired balance within 3 years of PD onset [2]. Nevertheless, postural instability is a feature of PD but occurs in later stages of PD and otherwise suggests an alternative diagnosis [51]. Based on these findings, PD patients being classified as in the early stage of PD should have a H&Y ≤ two and therefore no postural instability, whereas PD patients of advanced stage should have a H&Y > two. To strengthen the clear differentiation between early and advanced stages of PD, we added a criterion based on disease duration. The “honeymoon period” of PD is mostly accepted to last around 5 years after PD diagnosis, with sustained and satisfying symptomatic relief and minimal side effects by pharmacotherapy [52,53]. As there is no consensus on the key factors for diagnosing advanced PD, different approaches are discussed [54,55]. In the CEPA study, the majority of participants (up to 92%) considered disease duration as a determinant factor for the diagnosis of advanced PD, with a mean disease duration of 9.17 ± 1.95 years (median = 10.00 years) [56]. Therefore, our arbitrary definition of early PD stage as a less than 5-year disease duration and advanced PD stage as a more than a seven-year of disease duration should ensure a clear differentiation among the two groups.

The observation of reduced energy intake among advanced PD patients supports the idea that weight loss during the later stages of PD might be explained by a reduced energy intake and not by increased energy expenditure. This is in accordance with what other researchers have argued [26] and could be clinically important, since weight loss has been previously associated with worse health outcomes (e.g., early mortality) among PD patients [11]. Indeed, long-term reductions in energy intake among PD patients could lead to malnutrition, which has additional negative health consequences in addition to worsening the PD disease process [11]. It is important to note that the observation of lower energy intake contradicts previous findings based on patient self-reports that showed a higher energy intake among advanced PD patients vs. healthy controls [22]. This is not totally surprising, since self-reported energy intake data have been shown to have low validity across the field of energy intake estimation [27], potentially pointing towards biased reporting in the field of PD, where weight loss is a known clinical problem [11]. It should also be noted that this study was conducted in a controlled laboratory setting, while previous studies have been conducted in real life. This might affect the reported outcomes, together with potential differences in cultural eating behavior habits (e.g., in Germany in our case vs. Italy in [21]). Furthermore, cognitive impairment might further distort self-reported data among PD patients. On top of that, when reports are completed by caregivers instead of the patients themselves, further biases might be introduced. With that said, it would be interesting to conduct a longer duration objective study (e.g., in an inpatient setting similar to this study [57]), in order to validate our current findings. This is of special interest, since a lower energy intake among advanced PD patients can be regarded as an important clinical finding, with potential implications in PD monitoring and treatment. 

Overall, the PD-specific motor symptoms of UE tremor, subjective eating problems, dysphagia and a fewer number of spoonfuls could explain the observed reduced energy intake among the advanced PD patients according to our model. Targeted interventions for patients with greater tremor could potentially be helpful to improve energy balance in this group. Furthermore, interventions targeting the number of spoonfuls might also be clinically useful in order to prevent weight loss and malnutrition in PD patients. Such approaches could perhaps integrate feedback from widely available technology-assisted monitoring devices such as smartwatches, in order to train PD patients to balance their energy intake by increasing the number of spoonfuls during meals [58]. Current research in this domain such as [59,60] might allow for the development of automatic, user-friendly assistive everyday tools for monitoring and modifying plate-to-mouth movement in PD-patients. Indeed, such efforts already exist in other domains (e.g., in body weight management interventions [61]) and could be integrated with existing PD-specific assistive devices (e.g., [62]), or even home-centered support intervention platforms [63], designed to assist eating-related motor function in PD patients. The subjective eating problems, as well as dysphagia, are also important clinical targets that can be managed by health personnel in order to better balance energy intake. 

Surprisingly, we did not find an impact of taste and smell problems to explain the observed lower energy intake among advanced PD patients vs. early PD patients and healthy controls, although the included PD patients had significantly impaired olfaction, as is known in PD [6]. However, this is in line with studies investigating impaired olfaction as a cause of weight loss in PD patients that did not report a change in energy intake [64,65].

Our finding of lower water intake among advanced vs. healthy controls is in line with studies that have included self-reported data from PD patients [22,66]. This might be an important contributing factor to the high rates of constipation caused by PD pathology itself and being evident at all stages of PD [67]. Therefore, constipation management in PD should always mean to advise for sufficient hydration to complement medical interventions such as laxative drugs and dietetic supplementation in order to increase their limited effectiveness [68,69]. Another important problem regarding low water intake among PD patients is its potential impact on urinary tract infections [70]. As mortality from urinary tract infection is common in PD [42], mitigating dehydration might prove to be key to reducing this risk, which is similar to what has been found among groups of elderly people [70]. Overall, our results suggest that more attention should be paid to water intake among PD patients, potentially designing and evaluating treatment strategies targeting increased water intake.

With regard to the identified differences in meal component intake across the groups, the observed lower energy intake among advanced PD patients vs. both early PD patients and healthy controls could be explained by food texture with a lower intake of solid and semi-solid food among advanced PD patients, since there were no differences in the amount of consumed soft food. This observation is in line with subjective data obtained from PD patients in a previous study [66], where the authors suggested that the “avoidance” of solid food might be caused by eating problems and difficulties in the handling of food. Our data support this, since eating problems and dysphagia could partly explain the observed differences in energy intake in our study. Further, dysphagia is more prevalent among the advanced PD patients and might explain an enhanced preference for soft instead of solid food [5,71]. Therefore, both motor and non-motor symptoms of PD seem to have an impact on food choice that is dependent on food texture, with the avoidance of solid food, which is problematic from a nutritional perspective [66]. First, fruits and vegetables—foods with important health benefits—are commonly found in solid form. Secondly, chicken, fish and meat are usually eaten in solid format and provide important nutrients to support a balanced dietary intake. Therefore, advanced PD patients should get appropriate help from trained dieticians to find alternative ways to substitute nutritious foods that are commonly found in solid form in a soft format instead. 

Another interesting observation in our study was the high energy intake among all included groups of participants, PD patients and healthy controls. In previous studies, using comparable methodologies for meal monitoring in young adult women [72], group-level energy intake was approximately half of the one described here (e.g., ~300–400 kcal in previous studies vs. ~800 kcal in the current study). This could be partly explained by the high energy density of this study’s foods, but it should be noted that the amount of consumed food mass (grams of food eaten) was also high in comparison to our previous studies. It is worth mentioning that we served the participants with large food portions directly in the current study, while in past protocols the participants were allowed to define their own portion sizes from food trays with ad libitum food availability (e.g., [72]). This constitutes a protocol deviation that has been previously described as an important mediating factor for food intake [73] and should be taken into consideration when our current results are being considered. Furthermore, the current study was conducted in Germany, while our previous study was conducted in Sweden, and cultural eating habit differences might also partly explain the different lunch meal energy intakes. Since weight loss and weight gain have been observed during the PD process, portion sizes and energy density of the food that is consumed might be important intervention targets to better balance the diet of weight losing or weight gaining PD patients.

Our study has some important limitations. Since PD patients were assessed during the ON motor state, our results should not be generalized to patients’ behavior during the OFF motor state [31]. However, the observed differences between both early and advanced PD patients and healthy controls would probably have been greater in the case of patients eating their meal in the OFF motor state, taking into consideration the association between motor scores (UE tremor score) and eating problems, as well as dysphagia, which are worse in OFF stage. This points to the fact that reduced energy intake is a non-motor symptom in PD. Similarly, the observed differences would probably be even greater with the inclusion of patients with severe dysphagia and dementia, who were excluded from the current study due to ethical considerations [74]. In advanced PD patients, motor fluctuations with dyskinesia are important and dyskinesia is probably of relevance for long-term body weight development [65]. In the current study setting, dyskinesia was evaluated by AIMS just after the meal to ensure study performance in the ON motor state without severe disturbing dyskinesia. The extent of dyskinesia has not been investigated any further, as we do not assume any impact on energy intake during single lunch meals, especially as all three groups are comparable concerning BMI and the advanced PD group had a lower daily levodopa dose per kg body weight than the known doses in dyskinetic PD patients [75,76]. An additional limitation of our study is the imbalance in sex proportions between the included groups of participants. It is known that sex is an important factor in such settings, with males’ energy intake being higher than that of females [77]. The lack of recruitment availability of appropriate female PD patients led to the redesign of the analysis methodology post-data collection. The option to recruit additional patients was deemed unfeasible, due to the increased participant burden (especially among the advanced PD patients) that would be incompatible with the existing ethical approval. The updated analysis scheme allowed us to control for sex in all the performed analyses, mitigating potential bias due to the unbalanced proportion of females vs. males in the included groups, while satisfying appropriate power for the primary outcome analysis. Lastly, energy intake was studied during single lunch meals (e.g., a cross-sectional analysis), with relatively large food portions served to each participant, and should be taken into consideration when the study outcomes are considered. However, it is important to note that the study setting is directly reminiscent of everyday clinical food serving settings (e.g., a hospital lunch). Based on this, we argue that the observed lower energy intake in advanced PD patients is a clear indication that such patients might need greater support during their meals, in order to better balance their overall diet.

## 5. Conclusions

Advanced PD patients had a lower energy intake vs. early PD patients and healthy controls when controlling for sex. The number of spoonfuls during the meal, eating problems, dysphagia and UE tremor scores could explain most of the lower energy intake in advanced PD patients vs. healthy controls and the first three could explain ~50% between advanced vs. early PD patients. Reduced energy intake in the advanced PD stage seems to be a sign of disease progression within the course of PD and might be used as a surrogate marker. Furthermore, our results suggest that the energy intake side of the energy balance equation might be an important clinical target for PD patients who are at increased risk of weight loss and malnutrition (e.g., advanced PD patients). In the future, additional studies might need to consider updated research settings, incorporating the monitoring of repeated meals throughout the PD disease process, both in controlled and real-life environments, probably by using up-to-date, technology-assisted behavioral monitoring methodologies, e.g., [59,60]. Potential eating and diet-related changes should be evaluated further, as they occur across different stages of the disease, paving the way for the design and implementation of appropriate evidence-based interventions targeting the improvement of the nutritional status of PD patients, as well as improving their quality of life. 

## Figures and Tables

**Figure 1 nutrients-12-02109-f001:**
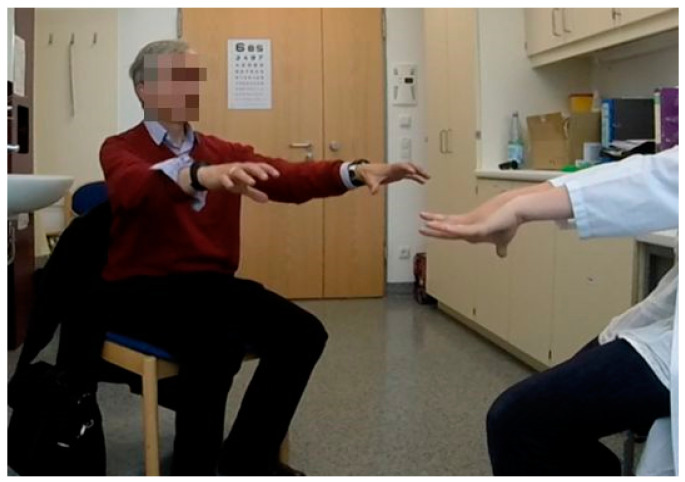
Standardized medical examination of upper extremity (UE) tremor.

**Figure 2 nutrients-12-02109-f002:**
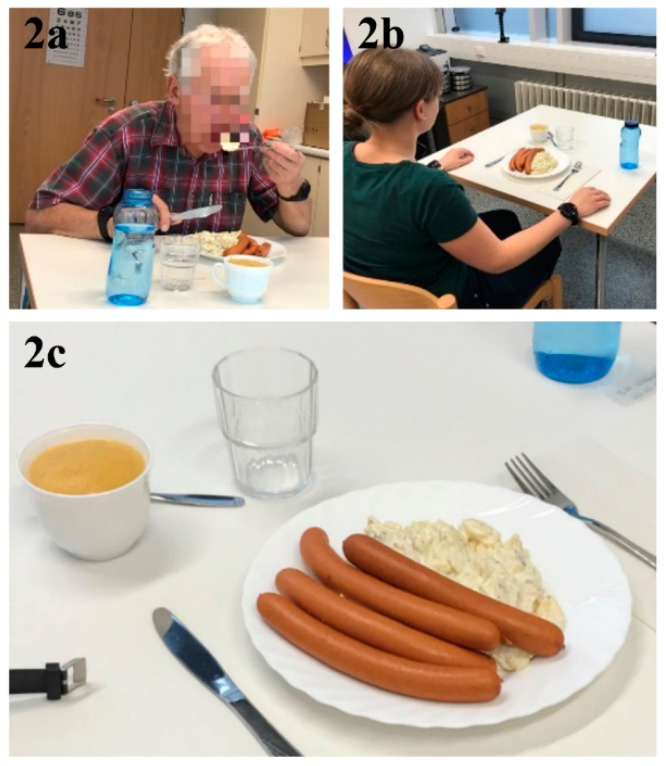
(**a**,**b**) The clinical lunch setting, and (**c**) the food that was served during the lunch meal (200 g sausages, 400 g potato salad, and 200 g apple purée).

**Figure 3 nutrients-12-02109-f003:**
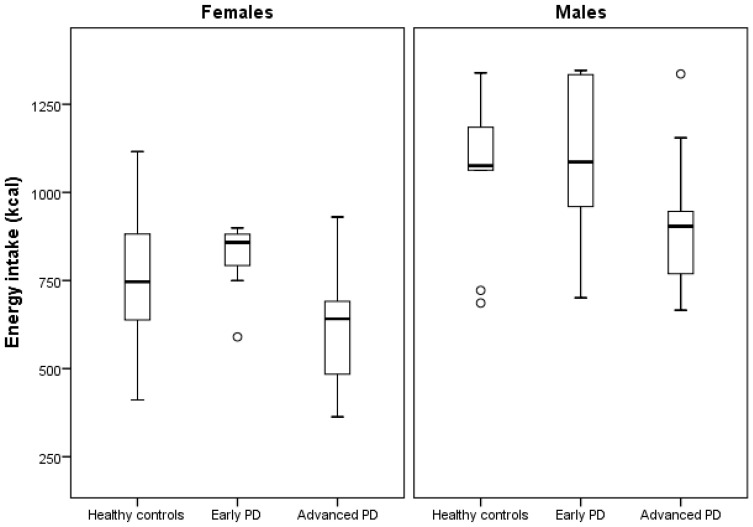
Boxplot illustrating group differences in energy intake (kcal) after sex stratification. Parkinson’s disease (PD). Each circle represents a participant with an energy intake value >1.5 × interquartile range.

**Figure 4 nutrients-12-02109-f004:**
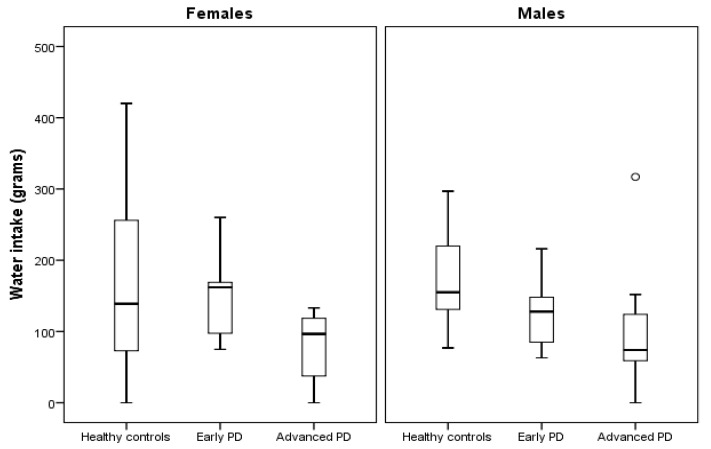
Boxplot illustrating group differences in water intake (grams) after sex stratification. Parkinson’s disease (PD). The circle represents a participant with a water intake value >1.5 × interquartile range.

**Table 1 nutrients-12-02109-t001:** Macronutrient composition and energy density of the foods (per 100 g) that were served during the standardized lunch.

	Potato Salad	Sausages	Apple Puree
Protein, g	2.0	14	<0.5
Carbohydrate, g	11	1.2	18
Sugars, g	1.2	<0.5	18
Fat, g	9.4	26	<0.5
Energy, kcal	140	295	84

**Table 2 nutrients-12-02109-t002:** Mean (standard deviation) characteristics of the three groups of (1) healthy controls, (2) early Parkinson’s disease (PD), and (3) advanced PD patients. Data are shown for females, males and total group sample.

	Healthy Controls			Early PD			Advanced PD		
	Females (*n* = 14)	Males (*n* = 9)	Total (*n* = 23)	Females (*n* = 7)	Males (*n* = 13)	Total (*n* = 20)	Females (*n* = 8)	Males (*n* = 13)	Total (*n* = 21)
**General characteristics**									
Age, years	61.2 (7.3)	64.4 (8.4)	62.5 (7.7)	62.4 (4.6)	59.9 (9.9)	60.8 (8.3)	64.0 (6.6)	64.0 (8.2)	64.0 (7.5)
Bodyweight, kg	70.3 (9.0)	87.0 (15.1)	76.8 (14.1)	79.5 (17.1)	86.2 (13.0)	83.8 (14.5)	76.6 (14.4)	85.8 (14.1)	82.3 (14.6)
Height, m	1.65 (0.06)	1.79 (0.09)	1.70 (0.10)	1.67 (0.06)	1.78 (0.06)	1.74 (0.08)	1.62 (0.06)	1.78 (0.08)	1.72 (0.11)
BMI, kg/m^2^	25.6 (3.4)	27.1 (3.6)	26.2 (3.5)	28.3 (6.9)	27.3 (4.3)	27.6 (5.2)	29.1 (5.4)	27.0 (3.5)	27.8 (4.3)
**PD Status**									
H&Y stage	1.4 (0.9)	1.3 (0.9)	1.3 (0.9)	1.9 (0.4)	2.0 (0.0)	2.0 (0.2) NA	2.5 (0.8)	2.2 (0.4)	2.3 (0.6) NA
Disease duration	0 (0)	0 (0)	0 (0)	3.6 (1.5)	2.9 (2.0)	3.1 (1.8) NA	10.9 (4.1)	12.9 (4.9)	12.1 (4.6) NA
**PD Medications**									
Levodopa Equivalent Daily Dose (mg/d)	0 (0)	0 (0)	0 (0)	431 (176)	578 (279)	527 (253) ^†^	1216 (525)	1059 (559)	1119 (538) *^,†^
Daily Levodopa dose (mg/d)/kg bodyweight	0 (0)	0 (0)	0 (0)	1.3 (1.7)	2.8 (2.6)	2.3 (2.4) ^†^	9.0 (4.5)	6.0 (3.0)	7.2 (3.9) *^,†^
Daily Levodopa Dose (mg/d)	0 (0)	0 (0)	0 (0)	107 (143)	245 (222)	197 (206) ^†^	669 (322)	527 (282)	581 (298) *^,†^
**Motor features of PD**									
Tremor UE	0.1 (0.3)	0.9 (1.1)	0.4 (0.8)	1.9 (1.6)	1.8 (1.6)	1.8 (1.5) ^†^	1.5 (1.2)	2.5 (1.5)	2.1 (1.4) ^†^
Brady/Hypokinesia UE	1.8 (1.4)	0.7 (1.0)	1.3 (1.3)	6.3 (3.5)	5.4 (2.0)	5.7 (2.6) ^†^	6.1 (2.5)	8.7 (4.0)	7.7 (3.6) *^,†^
Rigidity UE	0 (0)	0 (0)	0 (0)	2.0 (1.5)	2.3 (2.1)	2.2 (1.9) ^†^	1.3 (0.7)	1.5 (1.7)	1.4 (1.4) ^†^
**Non motor features of PD**									
Taste problems, N (%)	2 (14.3)	0 (0)	2 (8.7)	0 (0)	2 (15.4)	2 (10)	0 (0)	6 (46.2)	6 (28.6)
Smell problems, N (%)	3 (21.4)	2 (22.2)	5 (21.7)	6 (85.7)	13 (100)	19 (95) ^†^	7 (88.8)	13 (100)	20 (95.2) ^†^
Eating problems, N (%)	2 (14.3)	1 (11.1)	3 (13)	1 (14.3)	3 (23.1)	4 (20)	3 (37.5)	5 (41.7)	8 (40)
Dysphagia, N (%)	1 (7.1)	1 (11.1)	2 (8.7)	0 (0)	2 (15.4)	2 (10)	3 (37.5)	5 (38.5)	8 (38.1) ^†^
Has depression diagnosis, N (%)	5 (35.7)	2 (22)	7 (30.4)	2 (28.6)	3 (25.0)	5 (26.3)	2 (25.0)	3 (23.1)	5 (23.8)
Has constipation, N (%)	0 (0)	1 (11.1)	1 (4.5)	1 (14.3)	2 (20.0)	3 (17.6)	1 (14.3)	7 (53.8)	8 (40) ^†^

Parkinson’s disease (PD). Number of cases (*n*). Body mass index (BMI). Kilogram (kg). Meters (m). Hoehn and Yahr stage (H&Y). Milligram (mg). Upper extremity. (UE). * = significant difference vs. early PD patients when controlling for sex. ^†^ = significant difference vs. healthy controls when controlling for sex. Comparison not applicable due to the measure being part of the selection criteria (NA).

**Table 3 nutrients-12-02109-t003:** Model 1: Regression model comparing energy intake (kcal) among advanced stage PD patients vs. healthy controls as well as early stage PD patients vs. healthy controls when controlling for sex. Model 2: Regression model comparing energy intake (kcal) among advanced stage PD patients vs. early stage PD patients when controlling for sex. Two outliers were excluded (Cook’s distance greater than (4/n)).

Primary Outcome Models	B	t	*p*	Lower Bound 95% Confidence Interval for B	Upper Bound 95% Confidence Interval for B
1 *					
Sex	297.729	5.738	0.000	193.862	401.596
Early PD	40.651	0.651	0.517	−84.269	165.571
Advanced PD	−162.063	−2.623	0.011	−285.731	−38.394
2 *					
Sex	297.729	5.738	0.000	193.862	401.596
Healthy control	−40.651	−0.651	0.517	−165.571	84.269
Advanced PD	−202.713	−3.208	0.002	−329.214	−76.213

* = model is significant, *p* < 0.05. Sex: 1 = male, 0 = female. Parkinson’s disease (PD). Unstandardized b coefficients showing the change in kcal (B), the t test statistic (t), *p*-value (*p*).

**Table 4 nutrients-12-02109-t004:** Meal characteristics.

	Healthy Controls			Early PD			Advanced PD			
	Females (*n* = 14)	Males (*n* = 9)	Total (*n* = 23)	Females (*n* = 7)	Males (*n* = 13)	Total (*n* = 20)	Females (*n* = 8)	Males (*n* = 13)	Total (*n* = 21)
**Meal component intake**									
Potato salad intake (g)	214 (97)	320 (71)	255 (101)	209 (63)	339 (80)	291 (97)	161 (81)	269 (67)	232 (88) *^,†^
Sausage intake (g)	114 (25)	171 (42)	136 (42)	123 (26)	166 (40)	150 (41)	88 (21)	138 (38)	120 (41) *^,†^
Apple mash intake (g)	166 (54)	142 (74)	157 (63)	187 (21)	180 (35)	183 (30)	151 (46)	153 (46)	152 (45)
Water intake (g)	160 (113)	172 (69)	165 (97)	147 (63)	131 (49)	137 (54)	73 (48)	95 (85)	87 (73) ^†^
**Eating behaviors**									
Energy eating rate (kcal/min)	77 (21)	94 (17)	83 (21)	80 (12)	83 (26)	82 (22)	76 (23)	82 (20)	80 (21)
Eating rate (g/min)	49 (13)	55 (10)	51 (12)	51 (7)	51 (15)	51 (13)	49 (14)	50 (13)	50 (13)
Meal duration (min)	10.3 (2.1)	11.5 (2.3)	10.8 (2.2)	10.2 (0.6)	13.8 (4.7)	12.5 (4.1)	9.3 (3.2)	12.1 (4.7)	11.0 (4.4)
Number of spoonfuls	51 (12)	59 (13)	54 (13)	49 (7)	61 (17)	57 (15)	41 (8)	56 (17)	50 (16) *
Number of chews	543 (215)	686 (202)	599 (218)	475 (75)	903 (458)	753 (422)	448 (132)	828 (475)	683 (421)
Chews/g of food eaten	1.2 (0.7)	1.1 (.4)	1.2 (.6)	0.9 (0.3)	1.4 (0.6)	1.2 (0.5)	1.1 (0.5)	1.5 (0.8)	1.3 (0.7)

Parkinson’s disease (PD). Minute (min). * = significant difference vs. early PD patients when controlling for sex. ^†^ = significant difference vs. healthy controls when controlling for sex.

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
