# Peer review of "Lower Energy Intake among Advanced vs. Early Parkinson’s Disease Patients and Healthy Controls in a Clinical Lunch Setting: A Cross-Sectional Study"

_nutrients, 2020, doi:10.3390/nu12072109_

Round 1

Reviewer 1 Report

The manuscript entitled

‘Lower energy intake among advanced vs. early Parkinson disease patients and healthy controls in a clinical lunch setting: a cross-sectional study’ by Fagerberg  et al offers an objective assessment of the effect of deteriorating motor skills as contributing factors to weight loss in Parkinson’s disease. 

Review description of meal attributes

Abstract states

(400g sausages, 200g potato salad, 200g apple 25 purée and 500ml water)

Material and methods states 200g pre-heated sausages (solid food), 400g cold potato salad (semi-solid food), and 200g apple puree (soft food)

Table 1 Model one is described as early and late PD versus healthy control, yet B value implies early PD ate more calories than control.  Similarly model #2 which is described as advanced vs early describes healthy controls as calorie deficient. Are these correct?

In S3 the authors only provide data for significant differences and do not show the tremor UE score.  This should be included in S3.

Table S2, Smell problems rows the advanced and early PD data are in a different order to the preceding variables. 

Given that the discussion/conclusions of the analysis are predominantly based in the supplementary tables I think it would be prudent for this analysis to be in the main body of the paper.

The authors considered the affect of treatment on the effects they observed and in particular dyskinesia.  Did they also consider the different kinetics of treatment (ie advanced PD requiring more frequent dose of L-dopa or patients with duopa who’s on periods may differ to early onset patients.

Author Response

Point 1: “Review description of meal attributes

Abstract states (400g sausages, 200g potato salad, 200g apple 25 purée and 500ml water)

Material and methods states 200g pre-heated sausages (solid food), 400g cold potato salad (semi-solid food), and 200g apple puree (soft food)”

Response 1: The abstract has been corrected and now reads: “200g sausages, 400g potato salad, 200g apple purée and 500ml water”. Thank you for pointing this typo out.

Point 2: “Table 1 Model one is described as early and late PD versus healthy control, yet B value implies early PD ate more calories than control.  Similarly model #2 which is described as advanced vs early describes healthy controls as calorie deficient. Are these correct?”

Response 2: These are correct.

The comparison between advanced PD patients vs. healthy controls as well as the comparison between advanced PD patients vs. early PD patients were both significant (advanced PD patients had a lower kcal vs. both the other groups) while the comparison between early PD patients vs. healthy controls was not significant.

We have now added the following text to the results section in order to clarify this: “There was no significant difference in energy intake between early PD patients and healthy controls.”.

We did correct the numbering of the Table from “Table 1” to “Table 3”.

Point 3: “In S3 the authors only provide data for significant differences and do not show the tremor UE score.  This should be included in S3.”

Response 3: Tremor UE did not have “more than 10% effect on the primary outcome model” (i.e. the criteria for inclusion in the table) and was therefore not included in Table S3. We corrected the Table footnote that was erroneously referring to Tremor UE.

Point 4: “Table S2, Smell problems rows the advanced and early PD data are in a different order to the preceding variables. “

Response 4: We have now changed the order in accordance with your comment. Thank you for pointing this out.

Point 5: “Given that the discussion/conclusions of the analysis are predominantly based in the supplementary tables I think it would be prudent for this analysis to be in the main body of the paper.”

Response 5: The main discovery of our paper was the lower energy intake among advanced PD patients vs. both healthy controls and early PD patients. These comparisons are found in table 3 and are included in the results section of the main paper. Also, all the explanatory parameters and their values are included in Table 2. 

As for the supplementary tables, we already summarize the main outcomes from these in the text of the results section in our paper. We originally included the supplementary tables in the main paper as well. However, we decided to leave those in the supplementary material instead, since the paper already contain four tables and several figures, and we therefore thought that adding further tables would be more confusing rather than being helpful to the reader.

Point 6: “The authors considered the affect of treatment on the effects they observed and in particular dyskinesia. Did they also consider the different kinetics of treatment (ie advanced PD requiring more frequent dose of L-dopa or patients with duopa who’s on periods may differ to early onset patients.”

Response 6: Frequency of medications was not part of our current dataset. However, in our larger-scale, future studies, we are planning to expand upon medication effects on behavior and will pay a closer look at frequency of medication as well.

Reviewer 2 Report

This is an interesting article of a very important topic, not always well studied. 

I think is clearly developed but i would like to know:

  • Was any patient excluded or not finished the study?

Author Response

“This is an interesting article of a very important topic, not always well studied.

I think is clearly developed but i would like to know:

Point 1: Was any patient excluded or not finished the study?”

Thank you for your effort in reviewing our paper.

Response 1: None of the participants who started the study procedures were excluded from the study. However, two outliers were identified (Cooks distance >4/n) and the final regression analysis was completed with 62 participants (one early and one advanced stage PD patient were excluded from the regression models, see “Figure S1. Participant flowchart.” and the methods section of the main paper for more information).